# PhyloAug: Injecting Evolutionary information into GLMs via Data Augmentation *

## Abstract

Genomic Language Models (GLMs) suffer from the inherent problem of data scarcity, due to the cost, time and complexity of wet-lab experiments. Data Augmentation offers a solution; however traditional methods often disrupt the structural and functional properties of biological sequences. Furthermore, current GLMs struggle to capture evolutionary dynamics through standard data pipelines, limiting their understanding of nucleotide-wise importance and constraints. To address this, we present PhyloAug, a structure-aware, evolution-inspired augmentation method grounded in neutral theory. PhyloAug leverages Genomic Foundation Models (GFMs) to accurately perturb RNA sequences, guided by phylogenetic analysis via PAML to identify evolutionarily neutral site-wise positions where mutations are unlikely to affect function. These sites are concatenated with RNA secondary structures, ensuring that augmentations respect native structural constraints while embedding signals of neutral evolution. We further validate our method through a direct comparison of predicted neutral sites with Rfam-annotated conserved regions. We demonstrate that by enriching training data with these evolution-guided augmentations, PhyloAug improves GFMs on well-established RNA benchmark tasks, and further enables GFMs to internalise conserved sequence patterns and evolutionary constraints. We demonstrate this through by establishing a novel task requiring evolutionary reasoning, conserved site detection. PhyloAug demonstrates significant performance improvements of up to 12.9% MCC and 17.2% F1-Score across our key tasks.

## 1 Introduction

Genomic Foundation Models (GFMs) are large-scale machine learning models composed of millions to billions of parameters, pre-trained across an extensive corpus of genomic data. GFMs have been highly appraised for their adaptability to similar genomic tasks, with fine-tuning acting as the transfer layer to allow a general model, pre-trained through unsupervised learning, to be tuned for a specific task on a much smaller set of supervised data. A wave of innovation has recently demonstrated GFMs potential to decipher the language of genomics, DNA, RNA and proteins, with huge successes in protein structure modelling with ESM3 (Hayes et al., 2025), Evo 2 for DNA language modelling (Brixi et al., 2025), and the identification of new translation-associated motifs in plant RNA with PlantRNA-FM (Yu et al., 2024).

Although these advances highlight the power and potential of GFMs, their impact is constrained by fundamental limitations in both the data they rely on and the evolutionary understanding they capture. Prominent genomic benchmarks such as OmniGenBench (Yang et al., 2024) and BEACON (Ren et al., 2024) aim to provide curated, diverse genomic datasets to mitigate data scarcity, however these benchmarks must rely on biological laboratories to verify data, where

rigorous pre-processing and data validation techniques are applied with diverse sequencing technologies and wet-lab experimentation. Obtaining verified labels with biologically complex tasks, such as structural annotation (Watters et al., 2016) and functional annotation of long non-coding RNA (Chowdhary et al., 2021; Mattick et al., 2023a), requires significant fees and extensive biological expertise. GFMs are highly sought after to predict the outcome, as to minimise the time and costs associated, however without the original data to fine-tune the GFM, we are unable to obtain accurate results.

Whilst GFMs perform well on many tasks that rely on just an individual sequence-to-sequence mapping, their understanding of sequence evolution, how a biological sequence evolves over time and across species, has proven to be limited. This lack of evolutionary understanding has been demonstrated by three recent works (Albors et al., 2025; Ektefaie et al., 2025; Benegas et al., 2025), each discussing unique mitigation strategies. (Ektefaie et al., 2025) emphasises the importance of modelling sequences concurrently to develop an evolutionary understanding, and (Albors et al., 2025) and (Benegas et al., 2025) utilise carefully curated Multiple Sequence Alignment (MSA) and phylogenies to learn the evolutionary distinct signals between species.

Rather than propose new architectures, which demand substantial training resources and curated alignments, we introduce PhyloAug, a data augmentation framework that injects evolutionary information into existing GFMs. PhyloAug leverages neutral evolution and structural constraints to generate biologically faithful augmentations, and is readily applicable to non-coding RNA tasks. Whereas prior augmentation methods for coding RNA exploit codon amino-acid redundancy, such strategies do not transfer to non-coding RNA. PhyloAug addresses this key gap in genomic augmentation by providing a biologically grounded method tailored to non-coding RNA. We evaluate the effectiveness of our method on evolutionary tasks by introducing a novel evolution-aware task, conserved nucleotide prediction. We further validate general utility on a commonly known non-coding RNA task, RNA secondary structure prediction. Together, these contributions establish PhyloAug as a scalable and biologically grounded strategy to enhance evolutionary reasoning in GFMs.

## 2 Background & Related Work

### 2.1 Data Augmentation in Genomics

Data augmentation is a widely recognised field and has been applied in numerous areas across computer science (Trabucco et al., 2023; Mumuni and Mumuni, 2022; Li et al., 2022). However, genomics data is context-dependent (Lee et al., 2024), and thus widely applied techniques usually applied in Natural Language Processing, such as random substitution and input reversal, cannot easily be applied to genomics data (Sanabria et al., 2024). Furthermore, unlike traditional tasks such as sentence-modelling or image-based analysis, we as humans cannot accurately determine the label of genomics data merely by the predictive input, and must rely on biological wet-lab verification. Substitution of the real label with a synthetic label may violate a crucial assumption of the underlying data distribution, and the augmented data may not be supported by the true data distribution (Shao et al., 2022). Thus, we must preserve the original data labels during augmentation.

Previous genomic data augmentation methods focus on augmenting coding RNA, through methods such as synonymous mutation, as shown in EvoAug (Lee et al., 2023). As non-coding RNA does not contain codons, these methods cannot transfer to non-coding RNA, leaving a key research gap that PhyloAug seeks to fill. These key challenges of genomic modelling motivate our work, PhyloAug, a novel data augmentation methodology leveraging evolutionary biology to amplify the predictive power of genomic foundation models for non-coding RNA-specific tasks.

## 2.2 Mutations in Evolutionary Biology

PhyloAug is motivated through the widely renowned neutral theory within evolutionary biology. This theory asserts that the majority of evolutionary changes at the molecular level, within DNA, RNA and proteomic sequences, are the result of random genetic drift of neutral mutations, rather than Darwinian selection (Kimura, 1968). Whilst this theory is highly controversial within molecular biology (Jensen et al., 2019; Kern and Hahn, 2018), it is widely accepted that neutral mutations are a fundamental part of molecular biology. As neutral mutations often do not have observable phenotypic effects, they provide a biologically sound method to induce variation within training data through data augmentation without disrupting underlying functional signals.

Neutral mutations in coding regions, especially those that do not change the resulting protein, are well understood and often used as benchmarks in evolutionary studies, such as the McDonald–Kreitman test (Charlesworth and Eyre-Walker, 2008). However, while neutral changes also occur frequently in non-coding RNAs, it is much harder to differentiate between mutations that affect function and neutral mutations. This is because non-coding RNAs lack a corresponding amino-acid, thereby making it more difficult to detect the effects of mutations. Many non-coding RNAs, particularly long non-coding RNAs, are believed to evolve through nearly neutral processes (Mattick et al., 2023b), where most variants appear "noisy" due to their selective impact being too small to clearly distinguish from random drift. In practice, identifying functional sites in ncRNAs often requires using a combination of structural conservation, covariation patterns, and sequence conservation, rather than relying on simple sequence conservation as in coding RNA. In our work, we predict the neutral mutations using structural conservation estimated through Rfam covariance model-based alignments and the RNA secondary structure, and utilise PAML to identify sequence conservation patterns. We motivate PhyloAug as an augmentation methodology that aims to utilise these neutral mutations within augmentation. We integrate this evolutionary data by obtaining neutral site estimates through computationally-derived evolution.

## 2.3 RNA Structure in GFMs

As well as utilising evolutionary principles, we also aim to incorporate structural data within our pipeline, as to preserve RNA-Protein interactions and prevent the model from learning impeding sequences that may obscure the original function. Previous work such as OmniGenome (Yang et al., 2025) and RNAErnie (Wang et al., 2024) has demonstrated that incorporation of the RNA secondary structure can provide additional context, such as vital motifs within the RNA that must be preserved. Thus, by incorporating the secondary structure in our pipeline, we can identify key structural motifs important to function. Many RNAs released contain their secondary structure, however if the secondary structure cannot be obtained, we utilise ViennaRNA (Lorenz et al., 2011), a secondary structure prediction method based on thermodynamic principles, to estimate the true secondary structure. This guides our augmentation process to minimise disruption through avoiding computationally-derived mutations for the predicted secondary structure, preserving the original function.

## 2.4 Integrating Evolutionary Information within GLMs

Previous work primarily focuses on the usage of Multiple Sequence Alignment (MSA) or Phylogenetic information through Phylogenetic trees. MSA is used to align DNA or RNA sequences that are evolutionary similar, as to uncover the evolutionary mutations that have occurred per sequence. Phylogenetic trees are branching diagrams that illustrate the evolutionary relationship of a single or group of organisms. MSA is often termed as a "horizontal approach", and Phylogenetic Analysis as a "vertical approach", in which the evolutionary structure of the sequences is preserved (Merkl and Sterner, 2016), although just incorporating the phylogenetic tree is not enough to reach this vertical approach, and we must further utilise ancestral reconstruction tools such as PAML (Phylogenetic Analysis by Maximum Likelihood) (Yang, 2007). Ancestral

reconstruction re-creates the original sequences before evolutionary mutation occurred, thereby allowing us to make direct comparisons and accurately determine the types of mutations that occurred at each site of the nucleotide sequence. Whilst PAML is traditionally used for coding RNA, previous studies have demonstrated the use of the BaseML function for phylogenetic analysis of non-coding RNA (Hu et al., 2019). This provides a clear and accurate framework for the identification of neutral mutations by utilising the evolutionary context.

GFMs generally focus on the incorporation of MSA and phylogenetic information within the model pipeline, as a way to directly inject evolutionary information directly into the model. There have been several approaches, such as RNA-MSM (Zhang et al., 2023), a GFM pre-trained on RNA-MSA data, and the MSA-Transformer (Rao et al., 2021) proposed for Protein Language Models, enabling pre-training across a huge variety of MSA data. However, whilst incorporating MSA data has proved to be beneficial, GPN-MSA (Benegas et al., 2025) suggests the incorporation of evolutionary information alongside the MSA can further improve performance, as through aligning the MSA with the same gene across 100 vertebrate species, they achieve state-of-the-art performance in variant effect prediction. (Albors et al., 2025) and (Zhou et al., 2025) are further recent examples of integrating phylogenetic information within genomic models. CSFold establishes several key limitations of MSA, such as the reliance on the most common nucleotide, rather than establishing a clear evolutionary trend and pattern, and further utilises PAML and statistical tests to provide insight into the evolutionary trends of the data. PhyloGPN establishes a novel training paradigm where the training loss is used to model the evolution of aligned nucleotides given a phylogenetic tree, thus training the model to inherently understand nucleotide evolution.

### 2.5 OVERVIEW

These previous works have established that adding additional evolutionary-based information through MSA or Phylogenetic Trees may improve algorithm performance, however the method including this information varies greatly, and is algorithm-specific. We propose a one-size-fits-all solution for non-coding RNA, which can be applied to any gFM through fine-tuning, utilising the evolutionary and structural information through data augmentation.

## 3 METHODOLOGY

### 3.1 OVERALL PIPELINE

This section introduces our overall pre-processing pipeline to predict neutral sites within non-coding RNA. We begin by discussing the biological theories used within our pipeline to ensure that the nucleotide sites we mask minimally impact biological function or structure. Next, we dissect our pipeline and discuss each key section of our approach; gather homologous sequences using BLASTN and the nt database, utilise biological pipelines to establish neutral positions and mask these positions, recover masking percentage to a set threshold (if above) and the strategy used to perturb sequences with our GFM. One central limitation with this biological pipeline is that we do not consider the folded structure of the RNA, which may influence our mutations (a base-pair is less likely to mutate than an unpaired site). Thus to mitigate this, we discuss our methodology for integrating Rfam-annotated consensus structures and embedding the secondary structure into our GFM perturbations. Lastly, we discuss the establishment of a baseline comparison method, MSA-Only. With this complete pipeline, PhyloAug can augment non-coding RNA while adhering to the evolutionary and structural features of our data.

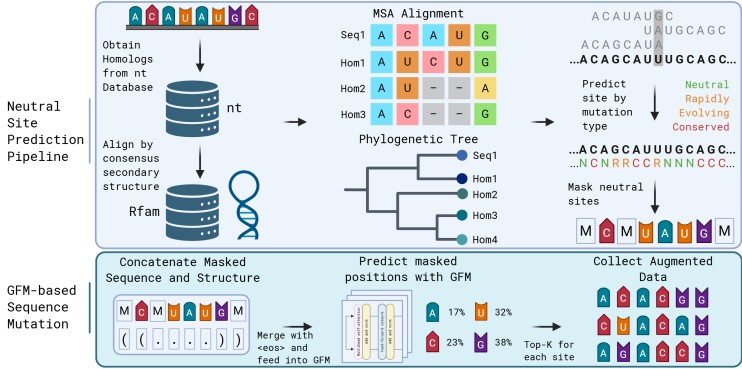

Figure 1: Overview of the augmentation pipeline. The process begins with the retrieval of homologous sequences from the NCBI nt database. Sequence-only homologs, homologs that do not belong to the same structural family, are filtered out using annotations from the Rfam database. A multiple sequence alignment is then constructed using MAFFT, and a phylogenetic tree is constructed using FastTree. These are then fed into PAML, where we employ ancestral sequence reconstruction and the estimation of site-specific evolutionary rates. Based on these rates, each site is classified as conserved, rapidly evolving, or putatively neutral. Only sites inferred to be neutral are masked. The masked sequence is then concatenated with its corresponding secondary structure and passed to the genomic foundation model, which perturbs the masked regions to generate the final augmented sequence set.

## 3.2 IDENTIFYING NEUTRAL MUTATIONS

A neutral mutation in is a change in a genomic sequence that has no effect on function or organism fitness. In coding RNA, synonymous substitutions (base changes that do not alter the amino-acid) are likely neutral mutations, as leaving the protein sequence unaffected will likely result in the function being unaffected also (Calderoni et al., 2016). Whilst identifying neutral mutations is a well known and explored field in the realm of coding RNA, for non-coding RNA (ncRNA), this problem is not so trivial. In coding RNA, a mutation can directly affect the amino-acid codons, thereby it is much easier to detect a direct change. Unlike coding RNA, ncRNAs do not have codons, and function through structural and regulatory roles, meaning mutations cannot be directly assessed through the amino-acid chain. Furthermore, mutations that do not alter the sequence function may still impact secondary structure, RNA-protein interactions, or expression. Therefore identifying neutral variation in ncRNA requires additional information, including the folded secondary RNA structure, and the usage of comparative genomics to identify nucleotides that are susceptible to change.

Our pipeline begins with the separation of each sequence from the training dataset, and an exhaustive homology search using NCBI's nt database, which contains over 116M RNA or DNA sequences[1]. This allows us to enrich the dataset by integrating evolutionary information from closely related sequences, thereby forming the basis of our Multiple Sequence Alignment (MSA). To ensure the sequences are homologically related, we utilise an e-value of $1e - 5$, and remove sequences that match within 5% of our original sequence identity to prevent duplicate sequences from obfuscating our analysis. After homolog collection, we utilise Rfam to further remove sequences that do not share the same structural family as the true sequence. A common issue with accurately identifying homologs is a close sequence identity but an invalid underlying structural family, thus this step removes any misaligned sequences. Thus, we have ensured our collected homologs are both sequentially and structurally aligned with our true sequence. Next,

---

[1]Whilst the nt database is huge, the homology search is batched to prevent long wait times

MAFFT (Katoh and Standley, 2013) is used to computationally align the sequences through MSA, accounting for evolutionary events such as insertions, deletions, and substitutions that have accumulated over extensive evolutionary timescales. We aim to measure the evolutionary distance between our homologs, providing our method an understanding of the cross-species relationships and the evolution of the sequence over time. However, phylogenetic analysis of MSA alone has many limitations, such as obscuring compensatory substitutions due to prioritising column-wise frequency over evolutionary information (Zhou et al., 2025).

In parallel to the MSA, we construct a phylogenetic tree based on the aligned RNA sequences within each homologous family using FASTTREE (Price et al., 2010). The selected phylogenetic analysis tool, PAML (Phylogenetic Analysis by Maximum Likelihood), is known to be inaccurate when constructing phylogenetic trees, thus FASTTREE is used to provide accurate phylogenies. The incorporation of a phylogenetic tree provides a framework for understanding the evolutionary trajectory of RNA families, revealing ancestral lineages and pinpointing evolutionary events obscured by MSA. To complete our phylogenetic analysis, we combine the aligned sequences and phylogenetic tree with PAML, to perform ancestral sequence reconstruction. This step utilises maximum likelihood methods to infer probable ancestral RNA sequences at internal nodes of the phylogenetic tree.

Rather than directly utilising reconstructed ancestral sequences, we opt to analyse the evolutionary patterns to rule out conserved and fast-evolving mutations. In particular, we employ PAML's baseml tool to estimate site-specific substitution rates from the provided MSA and phylogenetic tree. Sites that exhibit very low substitution rates are inferred to be conserved, likely due to structural or regulatory importance, while rapidly evolving sites may indicate natural selection or adaptation. Both types are assumed to be functionally important and are therefore excluded from our candidate set of neutral mutations, as is in-line with previous phylogenetic analysis of non-coding RNA (Meyer and von Haeseler, 2003; Knies et al., 2008).

To identify these constrained positions, we fit nucleotide substitution models with rate variation across sites using a discrete gamma distribution and empirical Bayes approaches. We obtain relative rate estimates for each site, and classify sites with posterior means $< 0.8$ as conserved, $> 1.2$ as fast-evolving, and those in between as neutral. These thresholds were chosen for their interpretability and robustness; they provide a symmetric margin around the neutral expectation (rate $\approx 1$) to accommodate for natural variation. Similar strategies were previously adopted in phylogenetic analyses of rate variation (Yang, 1994; 1996), where "conserved" and "accelerated" sites are defined relative to the neutral background rate. Importantly, these thresholds do not imply strict biological boundaries, but reduces the likelihood of perturbing functionally important sites during augmentation.

A maximum and minimum threshold of masked nucleotides must be set for each task, however the rates used are task-dependent, and many sequences can achieve upwards of 50% site-wise masking, or may have less than 5% of identified neutral sites. This may affect the stability and effectiveness of the generated augmentations, where site under-identification may result in insufficient sequence diversity, and over-identification increases the likelihood of damaging the original biological signals. To ensure reliability of our results, researchers should exclude sequences that fail to meet a minimum neutral sites, and reduce the theoretical maximum to prevent over-perturbing sequences.

### 3.3 Combining Neutral Positions with gFMs

Our masking strategy includes the consensus secondary structure from Rfam, however it does not consider the individualised RNA structure when identifying conserved or rapidly evolving sites. Providing the masked sequence alone could result in perturbations that change the underlying structure, which may render them invalid or harmful. To address this, we concatenate the true secondary structure label with the masked sequence to inject secondary context into the model.

It is for this reason that we selected OmniGenome as our GFM to perturb the RNA sequences, as OmniGenome was pre-trained with concatenated RNA secondary structures and sequences, and thereby was explicitly trained on the sequence-structure relationship. It should be noted that if the underlying RNA structure is not given within the dataset, we utilise ViennaRNA (Lorenz et al., 2011) to estimate the structure.

To accurately fill in the masked positions, we utilise a top-k approach, where nucleotides with a very small probabilistic rate ($< 5\%$) will be disregarded. This prevents our approach from choosing nucleotides that are very unlikely to occur through natural evolution, and those that may break the structural constraints. This allows our augmentation method the potential to generate huge amounts of augmentations, for example, a sequence with 15 masked positions has a theoretical maximum of $4^{15}$ perturbed sequences. Once the augmentations are complete for the training set, we merge the augmented sequences with the original dataset, and fine-tune our models.

### 3.4 MSA-ONLY BASED APPROACH

Notably in our pipeline, we split the type of mutation into three separate types, neutral, rapidly evolving and conserved. Through this description, it is possible to estimate these categories using MSA alone, although doing so is known to be unreliable. To prove the effectiveness of embedding phylogenetic analysis within our pipeline, we incorporate this as a comparator, as to provide further information on the importance of each part of the pipeline.

## 4 EXPERIMENTS

### 4.1 EXAMINING METHODOLOGY EFFECTIVENESS

Currently there is no established method to reliably extract the neutral sites within non-coding RNA, however Rfam (Ontiveros-Palacios et al., 2025) holds annotations for the conserved nucleotide sites within each RNA family. Thus, to investigate our method's ability to reliability and effectively circumnavigate conserved nucleotides within ncRNA, we compare our estimated conserved sites with the ground truth. To accomplish this, we first randomly select 52 diverse Rfam families, ensuring that each family contains at least 10 homologs to build an accurate conserved nucleotide space. To perform the experiment, we randomly extract a sequence from the homologs within the family, build our MSA by a BLAST search with the extracted sequence, and perform neutral site identification for our method. We measure the success of our method by calculating overlap between the conserved sites and our predicted neutral sites, where the conserved positions are obtained from the original Rfam-annotated data. To further establish the effectiveness of the incorporation of the Rfam-family alignment, we remove the Rfam part of our pipeline and show only the results of the phylogenetic analysis section of our method.

| Method | Average | Min | Max |
|---|---|---|---|
| PhyloAug | 0.047% | 0.029% | 0.073% |
| PhyloAug-No-Rfam | 0.067% | 0.054% | 0.081% |
| MSA | 0.114% | 0.102% | 0.134% |
| RANDOM | 0.126% | 0.117% | 0.151% |

Table 1: Average number of conserved nucleotides masked with each augmentation method. PhyloAug represents the full pipeline described in Overall Pipeline, PhyloAug-No-Rfam represents the pipeline without using Rfam to predict the conserved nucleotides, MSA represents the MSA-only methodology previously described, and Random represents masking based on purely random nucleotides with no constraints. A masking rate of 15% was used for each method.

Table 2: Model performance across conserved nucleotide and phylogenetic distance tasks with and without augmentation.

| Model | Cons Sites | | Aug Cons Sites | | No. Augs |
|---|---|---|---|---|---|
| | F1 | MCC | F1 | MCC | |
| SpliceBERT | $.724 \pm .17$ | $.578 \pm .15$ | $.802 \pm .10$ | $.687 \pm .07$ | 8 |
| HyenaDNA | $.633 \pm .16$ | $.275 \pm .12$ | $.692 \pm .14$ | $.358 \pm .11$ | 8 |
| RNA-FM | $.796 \pm .15$ | $.592 \pm .11$ | $.857 \pm .12$ | $.676 \pm .10$ | 8 |
| RNA-BERT | $.505 \pm .27$ | $.011 \pm .21$ | $.582 \pm .18$ | $.092 \pm .14$ | 8 |
| RNA-MSM | $.692 \pm .13$ | $.536 \pm .09$ | $.763 \pm .09$ | $.613 \pm .08$ | 8 |
| RNAErnie | $.617 \pm .19$ | $.252 \pm .12$ | $.668 \pm .12$ | $.324 \pm .10$ | 8 |
| OmniGenome | $.810 \pm .16$ | $.622 \pm .11$ | $.907 \pm .13$ | $.865 \pm .10$ | 8 |

Std across 3 random seeds reported in parentheses.

**Results**  We find that the complete PhyloAug pipeline, phylogenetic analysis and rfam-family alignment, performs best overall, with a clash rate of merely 0.047% with conserved nucleotides. All methods, even including our naive MSA method, outperforms random selection. We find that each part of our methodology increases effectiveness in circumnavigating conserved nucleotides, although the incorporation of the phylogenetic analysis alignment provides the most significant increase in effectiveness. These results show that PhyloAug produces biologically faithful augmentations by avoiding sites vital to function and structure.

### 4.2 Conserved Site Prediction

Building on this, we next test whether these biologically grounded augmentations actually improve model performance on evolutionary tasks. We first establish a dataset targeting conserved sequence features, we utilised RNA family MSAs collected from the Rfam database (Ontiveros-Palacios et al., 2025). Full family alignments were downloaded in Stockholm format, and conserved nucleotides were extracted. Alignments shorter than 50 positions were excluded to ensure sufficient sequence context for conservation analysis. Specifically, we focus on the models understanding of evolutionarily conserved nucleotide positions.

**Results**  Our results demonstrate improved performance across all models, although the degree of the improvement is largely model-dependent. OmniGenome and RNA-FM achieve the strongest results overall, and despite having strong performance, PhyloAug still results in a significant improvement. The largest relative improvements occur in weaker baselines such as RNA-BERT, showing that augmentation helps especially when models struggle to capture evolutionary constraints. Likely OmniGenome performs best due to its pre-training task with both secondary structures and sequences combined, thereby learning the structurally conserved nucleotides within pre-training. All models also show reduced variance across random seeds, suggesting that augmentation stabilises training.

### 4.3 Structural Prediction

**Experimental Design**  Finally, we evaluate whether the benefits extend beyond conservation tasks to RNA structural prediction. We utilise the three standard structural prediction datasets, Archive2, bpRNA, and rnastralign, as consistent with previous analysis and benchmarking methods. To evaluate the performance, we utilise the standard F1-Score, and combine it with Matthew's Correlation Coefficient (MCC), as to further evaluate the robustness of our model performance. F1-Score does not evaluate true negatives, thus by including MCC, we also evaluate the negative prediction aspect of our models.

Table 3: Raw Performance improvements (absolute gain over baseline) across datasets with augmentation.

| Model | Archive2 | | bpRNA | | StrAlign | |
|---|---|---|---|---|---|---|
| | F1 | MCC | F1 | MCC | F1 | MCC |
| SpliceBERT | .044±.016 | .066±.024 | .074±.025 | .065±.020 | .008±.003 | .012±.005 |
| HyenaDNA | .049±.022 | .075±.041 | .047±.030 | .096±.062 | .013±.004 | .020±.007 |
| RNA-FM | .025±.011 | .037±.016 | .023±.010 | .059±.027 | .008±.003 | .012±.005 |
| RNA-BERT | .364±.057 | .500±.071 | .064±.027 | .077±.031 | .022±.009 | .033±.012 |
| RNA-MSM | .058±.019 | .089±.027 | .115±.041 | .166±.055 | .015±.005 | .023±.009 |
| RNAErnie | .006±.016 | .010±.018 | .004±.006 | .006±.008 | -.002±.004 | -.002±.005 |
| OmniGenome | .007±.007 | .012±.012 | .029±.0017 | .044±.020 | .001±.002 | .001±.002 |

**Results** When testing our augmentation method for non-coding RNA, we find that for all structural prediction tasks, our augmentation methodology improves performance consistently across all models. For Archive2, we find that small models, such as RNA-BERT, which were previously unable to generalise to the sparse dataset has a major increase in both MCC and F1-score. Models with stronger performance, such as RNA-FM and RNA-MSM see a small increase F1-Score, but a comparatively larger increase in MCC, suggesting a reduction in false positives and negatives across model performance. Therefore, augmentations increase the robustness of the models, as well as their predictive accuracy. We see a similar trend in the bpRNA dataset, whereas rnastralign also shows signs of this trend, for top performing models such as RNA-FM and OmniGenome, there is little change in model performance. This is not unexpected however, as the model performance is almost perfect, suggesting that this may be the peak of the model understanding for this dataset.

## 5 Conclusions

In this work, we introduced PhyloAug, a data augmentation method incorporating both evolutionary and structural information to improve Genomic Foundation Models on downstream tasks. We established the importance of the key sections of our pipeline through an empirical experiment aligned with Rfam conserved nucleotides. We empirically demonstrated the effectiveness of incorporating evolutionary information into the data augmentation process with two key tasks, conserved nucleotide identification and sequence distance classification. Comprehensive experiments for non-coding secondary structure prediction illustrates the effectiveness of PhyloAug for general non-coding tasks as well as evolutionary-based. Future work may aim to integrate neutral evolution with batched sequences, to infer evolutionary important sites directly. Notably, we release our code using the publicly available GitHub repo: https://anonymous.4open.science/r/PhyloAug

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

# 6 Appendix

## 6.1 Sequence pattern preservation and distribution

In this section we aim to answer how effective our augmentation methodology is at preserving evolutionarily conserved contexts encoded within biological sequences. To achieve this, we utilise the non-coding RNA structural prediction datasets, and assess how closely the augmentations can reproduce the underlying characteristics of the set of homologous sequences. We perform this investigation on each individual RNA within all three ncRNA structural prediction datasets, as to provide a complete and comprehensive evaluation.

### 6.1.1 Experimental Design.

Traditional analysis of RNA motif preservation utilises a simple visual inspection of sequence logos, however due to the large amount of augmented sequences, we cannot show an accurate visual diagram across all data. To preserve the motif-specific interactions across each original RNA and it's augmented set, we compare the MSA-aligned homologs obtained for each RNA with the augmented set of RNA sequences. To fully understand the overall similarity across the entire training set, we average the Jensen-Shannon Distance (JSD) and Cosine Similarity scores across all augmented sequences as opposed with the MSA-aligned homologs. This gives us an average for how closely the augmented sequence is able to represent the underlying nucleotide distribution across the underlying data proportions. JSD is used to measure the similarity of the nucleotide distributions, and Cosine Similarity measures the overall pattern/shape of the nucleotide frequencies.

### 6.1.2 Results

Our results across all three datasets show a significant improvement for PhyloAug as opposed to MSA-only and random masking. We find that for each level of augmentation, we gradually approach the ground truth, and with only one level of augmentation, the overall result is the largest distance away from the ground truth. This is intuitive as our augmented sequences should represent similar homologs our MSA-aligned data. This further demonstrates the usefulness of a large set of augmentations, as with increasing augmentations, we draw closer to the ground truth. We find that the full PhyloAug pipeline results in the closest evolutionary distance from the ground truth, with random masking being the furthest. There is a significant difference between PhyloAug and the alternative methods, of which the gap is maintained as the augmentation level rises. This thereby proves our empirical result, being that the more augmentations, the better overall performance, with reducing returns. The low JSD values demonstrate that the nucleotide distributions of our augmented sequences are closely aligned with the original MSA homologs, and our high cosine similarity shows a similar overall pattern/shape of nucleotides. We thereby demonstrate empirically through performance and evolutionary distance the effectiveness of our method.

Table 4: JSD and Cosine Similarity results for three datasets

| | | Phylo Masking | | MSA-only Masking | | Random Masking | |
|---|---|---|---|---|---|---|---|
| Dataset / Augs | | JSD | Cosine | JSD | Cosine | JSD | Cosine |
| | 1 | 0.2431 | 0.7654 | 0.2740 | 0.7349 | 0.2896 | 0.7205 |
| | 2 | 0.2422 | 0.7674 | 0.2732 | 0.7361 | 0.2810 | 0.7291 |
| **Archive2** | 4 | 0.2414 | 0.7690 | 0.2728 | 0.7369 | 0.2745 | 0.7350 |
| | 8 | 0.2410 | 0.7694 | 0.2726 | 0.7373 | 0.2711 | 0.7379 |
| | 12 | 0.2402 | 0.7702 | 0.2726 | 0.7375 | 0.2703 | 0.7386 |
| | 1 | 0.2279 | 0.7799 | 0.2558 | 0.7499 | 0.2883 | 0.7202 |
| | 2 | 0.2264 | 0.7821 | 0.2550 | 0.7533 | 0.2780 | 0.7303 |
| **bpRNA** | 4 | 0.2242 | 0.7864 | 0.2544 | 0.7448 | 0.2708 | 0.7360 |
| | 8 | 0.2236 | 0.7879 | 0.2540 | 0.7456 | 0.2667 | 0.7393 |
| | 12 | 0.2228 | 0.7887 | 0.2536 | 0.7466 | 0.2653 | 0.7404 |
| | 1 | 0.2363 | 0.7838 | 0.2582 | 0.7619 | 0.2665 | 0.7577 |
| | 2 | 0.2362 | 0.7847 | 0.2564 | 0.7658 | 0.2613 | 0.7634 |
| **rnastralign** | 4 | 0.2358 | 0.7855 | 0.2548 | 0.7677 | 0.2576 | 0.7666 |
| | 8 | 0.2353 | 0.7861 | 0.2540 | 0.7689 | 0.2555 | 0.7685 |
| | 12 | 0.2346 | 0.7863 | 0.2536 | 0.7694 | 0.2549 | 0.7689 |