# OpenReview forum: "PhyloAug: Injecting Evolutionary information into GLMs via Data Augmentation"
_ICLR.cc/2026/Conference — ICLR 2026 Conference Withdrawn Submission_

### Official Review · Reviewer_z8Qj · 2025-10-30

**Soundness:** 2
**Presentation:** 1
**Contribution:** 2
**Rating:** 2
**Confidence:** 4

**Summary:**

The authors proposed a new strategy for training data augmentation for genomic language models. Overall, I find the idea of data augmentation guided by evolutionary and RNA structure constraints to be interesting and promising. However, the methodology was described rather obscurely, and the design choices are not well justified. Also, the evaluations are limited and do not convincingly demonstrate the benefit of the proposed strategy. I therefore recommend rejection in its current form.

**Strengths:**

1. The authors propose a novel and interesting idea of training data augmentation for GLMs based on RNA structures and MSAs.
2. There is sufficient discussion about the background and the trajectory of the methodology.

**Weaknesses:**

1. There is an issue with the overall framing of this work. There are mixed uses of GLM and GFM. I suggest unifying them into one. Also, it appears this work actually only studies RNAs. The terminology could be changed to RNA LM. In general, this work should be framed as a training data augmentation strategy for training RNA LM, or gLMs focused on RNA tasks.

2. How are the augmented data used to train or fine-tune the model? I find this important information to be largely missing. The authors should have provided the detailed training or fine-tuning setup for each model with the augmented data.

3. The method description could be improved. For obtaining the neutral sites, there are many steps, going from MSA construction to tree construction to rate estimation. There should at least be some descriptive statistics of the data obtained at each step.

4. The RNA structure and MSA were only used to select the sites for perturbation, and then the GLM OmniGenome was used to generate new data. This step is to some extent more important, but was not emphasized in the paper. This design choice should be justified by comparison with baselines, such as using other GLMs, random sampling, or sampling according to MSA.

6. The evaluation is limited to a handful of simple tasks. Consider using more comprehensive and biologically meaningful benchmarks, for example, BEACON (https://arxiv.org/abs/2406.10391). Besides, the conserved site prediction seems trivial since the same data were used in augmentation. The authors should have made it clear if it is a held-out test set. In structure prediction, it would be better to show the baseline performance or relative performance gain, rather than only the absolute gain.

**Questions:**

Overall, the writing could be improved. There are interspersed grammar errors. I find that statements in sections 1 and 2 are sometimes repetitive and not very coherent to read.

---

### Official Review · Reviewer_BxkT · 2025-10-31

**Soundness:** 1
**Presentation:** 2
**Contribution:** 2
**Rating:** 2
**Confidence:** 4

**Summary:**

PhyloAug proposes a method to augment non-coding RNA datasets to increase sequence diversity without the need to reassign labels. To accomplish this, PhyloAug performs a multiple sequence alignment on homologs obtained from the NCBI BLAST portal and then identifies sites on the original sequence that are under neutral evolution. These sites, denoted with a mask token, are selected as candidates for nucleotide permutation and are concatenated with RNA secondary structure predictions from Rfam or ViennaRNA to incorporate structural context. These two tracks are then fed into a biological sequence foundation model to predict possible nucleotides, and a top-k algorithm is used to select augmented sequences from these predictions. The authors demonstrate that the masked sites correspond well with Rfam conservation annotations and conserved nucleotides. Finally, the PhyloAug pipeline was used to augment foundation models evaluated on RNA secondary structure prediction benchmarks.

**Strengths:**

- Apart from terminology issues noted below, the background section provided a strong overview of the biological motivations of the method and relevant methods in the space.
- PhyloAug tackles RNA secondary structure prediction, which is a highly relevant and open problem in biological sequence modelling.
- PhyloAug shows strong performance benefits for smaller RNA foundation models.

**Weaknesses:**

- Although PhyloAug is elegant and biologically grounded, its empirical results are somewhat limited. The best-performing models in Section 4.3 do not consistently benefit from PhyloAug to a significant degree. Furthermore, the evaluation does not include SOTA models such as Evo2 or AIDO.RNA, which could further demonstrate the trend of modern models diminishing the need for such augmentation strategies.

- The methodology for training the models in Section 4.3 is unclear, which limits the ability to assess the validity of the results. Unless I have missed it, the manuscript does not report the training procedures or hyperparameters used for each model.

- There is insufficient detail in the manuscript for reproducibility. For example, it is not reported what the temperature setting is when ViennaRNA is used to predict secondary structure. This becomes important in downstream task evaluation on bpRNA or StrAlign, which contains a very diverse mixture of organisms that may require different temperature parameters for accurate RNA secondary structures. Also see the point below regarding the data-splitting strategy. What are the stds in 4.3 computed over: random data splits, or random seed during augmentation? These descriptors all affect reproducibility.

- As the authors have noted, the inclusion of evolutionary information into DNA and RNA modelling is well established. As such, there should be greater clarity on the differentiation of PhyloAug from existing methods, other than a focus on ncRNA and increased empirical performance. As it stands, PhyloAug appears to be a well-executed pipeline for RNA fine-tuning suited for a bioinformatics venue, rather than a manuscript with generalizable insights for the ICLR community.

Minor clarity points:

- There are several writing ambiguities that cause confusion at first glance. For example, on line 330, it's not immediately clear whether "remove the Rfam part of our pipeline" refers to the homology filter or the structural concatenation, and it's only after reasoning through the experiment that the ambiguity is resolved.

- The terminology used in the paper requires increased precision. It is incorrect to use "Genomic Language Model" as an umbrella term for both DNA and RNA modelling. While this is a minor point and does not contribute to my overall assessment of the paper, many passages do not make sense from a computational biology perspective. As one example, in the introduction, the line "Prominent genomic benchmarks ... BEACON (Ren et al., 2024)" is problematic because BEACON only contains RNA tasks, and "genomic" is understood to refer to DNA modelling. The current manuscript excludes comparison against actual DNA benchmarks such as the GUE, DGEB, DartEval, TraitGym, and only assesses RNA tasks. As I believe the author's intention is to focus on RNA, I suggest simply refraining from using "genomic" as a catch-all phrase for nucleotide sequence modelling and restricting the scope of the claims to RNA language modelling.

**Questions:**

- Its hard to assess the conclusions of Section 4.3 without further details on experimental setup. How was the data splitting conducted? I'm aware bpRNA has standard splits, but it seems important to report on the rest. Given the high amount of homology, one critical concern is data leakage. Since the sequences contained in bpRNA, StrAlign, etc. likely appear in BLAST, it seems to be a distinct possibility  that the PhyloAug method is essentially reducing the search space of augmentations to those that are likely to appear in the test set via family filtering and masking. I would appreciate the authors ruling out this possibility by either describing their data splitting, or running experiments to prevent this type of leakage.

- A minor claim of the paper from the introduction seems to be PhyloAug's efficiency in terms of training resources. However, from my understanding, running BLAST and MSAs takes substantial resources, and reporting the computational cost would help readers understand the practicality of this approach.

- The premise of section 4.1 hinges on Rfam conservation annotations being a reliable source of ground truth. However, if these annotations are themselves derived from a predictive pipeline, the evaluation primarily reflects how closely one computational model reproduces another, rather than how well PhyloAug captures biological reality. Additional details on how the Rfam conservation annotations were generated would be helpful.

- Minor suggestion, but including a few key equations, for example describing the core data flow shown in Figure 1, could be helpful for a computational audience.

- An interesting ablation could be to see how using ViennaRNA predicted vs Rfam structural annotations affects the performance of PhyloAug.

---

### Official Review · Reviewer_8EU8 · 2025-10-31

**Soundness:** 3
**Presentation:** 3
**Contribution:** 3
**Rating:** 6
**Confidence:** 3

**Summary:**

This paper proposes PhyloAug, a data augmentation framework for non-coding RNA that addresses data scarcity and the problem of traditional augmentation methods disrupting function. Based on "neutral theory," the method uses a complex bioinformatics pipeline to identify "neutral sites" with low functional impact . PhyloAug then masks these sites, concatenates the sequence with its secondary structure, and uses a GFM to fill the masked positions, generating new, biologically meaningful data . Experiments show the method effectively avoids disrupting conserved sites and improves GFM performance on ncRNA structure prediction and a novel "conserved site detection" task.

**Strengths:**

- Fills a Key Gap: Clearly addresses the important problem of lacking effective, function-preserving augmentation methods for ncRNA .

- Biologically-Grounded: The method is based on neutral evolution theory  and combines phylogenetic analysis with secondary structure to ensure the augmentation is reasonable.

- Method Validation: The experimental design in Table 1 is excellent . By calculating the overlap with Rfam-annotated conserved sites, it strongly demonstrates that PhyloAug effectively avoids disrupting functional sites (only 0.047% overlap).

- Performance Gains: Experiments show PhyloAug improves GFM performance on standard ncRNA structure prediction tasks and on a new evolutionary reasoning task (conserved site detection) .

**Weaknesses:**

- Contradictory Results: The F1 score in Table 2 (conserved site prediction task) decreases after augmentation. This strongly contradicts the abstract's claim of a significant F1 score increase ("17.2% F1-Score") and the text's claim that "all models show improved performance".

- Highly Complex Pipeline: The entire pipeline involves BLAST, Rfam, MAFFT, FastTree, PAML, and GFM inference , making it computationally expensive and difficult to reproduce.

- GFM Dependency: The perturbation step explicitly selects OmniGenome because it was pre-trained with structural information. This limits the generality of the augmentation method.

**Questions:**

- Table 2 Contradiction: Please clarify the contradiction between the F1 score drop in Table 2 and the F1 score increase claimed in the abstract. Which experiment does the "17.2% F1" increase in the abstract refer to?

- GFM Dependency: If a GFM not pre-trained on structure (e.g., RNA-FM) is used for the masked filling (perturbation) step, is PhyloAug still effective?

- "Neutral" Threshold Choice: Defining the "neutral" rate as being between 0.8 and 1.2  seems somewhat arbitrary. How sensitive is the downstream task performance to this threshold choice?

- Computational Cost: What is the average computational time required to run the full PhyloAug pipeline (from BLAST to GFM perturbation) to augment a single sequence?

---

### Official Review · Reviewer_j2b4 · 2025-11-01

**Soundness:** 2
**Presentation:** 3
**Contribution:** 3
**Rating:** 2
**Confidence:** 2

**Summary:**

This paper proposes a data augmentation mechanism specifically aimed at improving the performance of Genomic Foundation Models (GFMs) on non-coding RNA tasks. At the highest level, the PhyloAug pipeline consists of two main steps: site-rate estimation to identify (non-)conserved sites which are then masked, and a structure-conditioned GFM to impute the masked sites, thus leading to an augmented dataset. Using this procedure broadly improves performance on conservation and structure prediction tasks.

**Strengths:**

- Leveraging evolutionary information (by means of site-rate estimates) and structural information (by means of a structure-conditioned GFM) is an appealing way to bootstrap on top of existing procedures to produce reasonable data augmentations.

**Weaknesses:**

- The method uses secondary structures as part of its pipeline, and then also in the benchmarking tasks as ground truth. As far as I could tell, there is little to no discussion about the risks of test set leakage and ways in which this was avoided or mitigated. How do we know the test structures were never used during training? Was some careful train/test split employed?
- No insight is provided into the improved performance. For example, RNA-BERT's performance improves quite a bit with the augmentation. Any insights on this? Only aggregate metrics are shown. More detailed analysis would be good to rule out test set leakage.
- If the authors can argue that their benchmarks do not suffer from test set leakage -- such that performance can be attributed to the methodological advances and not other artifacts -- then the work becomes more attractive. Currently, there is limited discussion on the risks of test set leakage and the measures that have been taken to avoid or mitigate this. Train/test splitting is a fundamental (and challenging) task in biological settings.

**Questions:**

- Why do you use PAML at all? Doesn't FastTree already produce site rate estimates? I am not sure why you need ancestral sequences to estimate site rates.
- Would the method also work for coding RNA?
- Test set data leakage: how do you know this is not an issue in your benchmarks? Structures (and sequence data more broadly) are used in both training and testing. How do the training sequences and structures in training compare to testing? Are there exact or approximate overlaps?
- Why does RNA-BERT do so much better on Archive2 (Table 3)? Where is the improvement coming from? Explanation should rule out test set leakage.

---

### Note · Authors · 2025-11-28

**Comment:**

Thank you to the reviewers for their valuable feedback.

**Withdrawal Confirmation:**

I have read and agree with the venue's withdrawal policy on behalf of myself and my co-authors.